# The Economic Burden of Severe Acute Malnutrition with Complications: A Cost Analysis for Inpatient Children Aged 6 to 59 Months in Northern Senegal

**DOI:** 10.3390/nu16142192

**Published:** 2024-07-10

**Authors:** Bibata Wassonguema, Dieynaba S. N’Diaye, Morgane Michel, Laure Ngabirano, Severine Frison, Matar Ba, Françoise Siroma, Antonio V. Brizuela, Martine Audibert, Karine Chevreul

**Affiliations:** 1Research Unit, Expertise & Advocacy Department, Action Contre la Faim (ACF), 93100 Montreuil, France; 2ECEVE, UMR 1123, Université Paris Cité, Inserm, 75010 Paris, France; 3Unité D’épidémiologique Clinique, Hôpital Robert Debré, Assistance Publique-Hôpitaux de Paris, 75019 Paris, France; 4Action Contre la Faim, Dakar 29621, Senegal; 5Action against Hunger, 28002 Madrid, Spain; 6Centre d’Études et de Recherches sur le Développement International (CERDI) CNRS-IRD-UCA, 63000 Clermont-Ferrand, France

**Keywords:** economic burden, cost analysis, severe acute malnutrition with complications, Northern Senegal, societal cost, health economics, cost drivers

## Abstract

Severe acute malnutrition (SAM) is a high-fatality condition that affected 13.7 million children under five years of age worldwide in 2022, with complicated cases requiring extensive inpatient stay with an accompanying caregiver. Our objective was to assess the costs of inpatient treatment for complicated SAM in children aged 6 to 59 months in Northern Senegal and identify cost predictors. We performed a retrospective cost analysis, including 140 children hospitalized from January to December 2020 in five SAM inpatient treatment facilities. We adopted a societal perspective, including direct medical and non-medical costs and indirect costs. We extracted patients’ sociodemographic and clinical data from medical records and conducted semi-structured interviews with healthcare staff to capture information on time allocation and care management. A multivariable generalized linear model with gamma family and a log link was used to investigate the factors associated with direct costs. Costs are expressed in 2020 international USD using purchasing power parity. Mean length of stay was 5.3 (SD = 3.2) days and diarrhoea was the cause of the admission in 55.7% of cases. Mean total cost was USD 431.9 (SD = 203.9), with personnel being the largest cost item (33% of the total). Households’ out-of-pocket expenses represented 45.3% of total costs and amounted to USD 195.6 (SD = 103.6). Costs were significantly associated with gender (20.3% lower in boys), diarrhoea (27% increase), anaemia (49.4% increase), inpatient death (44.9% decrease), and type of facility (26% higher in hospitals vs. health centre). Our study highlights the financial burden of complicated SAM in Senegal in particular for families. This underscores the need for tailored prevention and social policies to protect families from the disease’s financial burden and improve treatment adherence, both in Senegal and similar contexts.

## 1. Introduction

Severe acute malnutrition (SAM) prevalence reached 2.1% in 2022 worldwide [1], representing 13.7 million children under five. That same year, across nine countries in the Sahel (Burkina Faso, Northern Cameroon, Chad, The Gambia, Mauritania, Mali, Niger, Northern Nigeria, and Senegal), 1.9 million children were admitted to health facilities for SAM, with 8.6% of them experiencing medical complications [2]. In Senegal, despite the country being one of the best performing in terms of nutrition indicators in the region, the prevalence of acute malnutrition remains high (2.8% among children under five in 2022), and regional disparities persist. The northern regions are the most affected and as such, acute malnutrition remains a public health concern in these regions [3]. As recommended by the World Health Organization (WHO) and the United Nations International Children’s Emergency Fund (UNICEF), the Senegalese national protocol for the management of SAM recommends treating children under five with uncomplicated SAM in outpatient programs (OTPs). Complicated cases require an extensive inpatient stay with an accompanying caregiver [4].

Previous studies in West Africa have shown that 19.2% to 43% of children with SAM required hospitalization [5,6] due to complications such as lack of appetite (anorexia) and/or bilateral oedema, diarrhoea, dehydration, acute respiratory infections, anaemia and hyperthermia. In Senegal, those children are hospitalized in Centres for Rehabilitation and Nutritional Education (CREN), i.e., specialized units hosted either in hospitals or health centres [4], which are two different levels of the Senegalese health system pyramid.

The objective of SAM inpatient management is to treat medical complications alongside the nutritional treatment, following the national protocol. Children are considered successfully treated if their clinical condition has been stabilized (appetite regained, complications treated). They are then referred back to OTPs to complete the ambulatory nutritional treatment for uncomplicated SAM. 

Previous economic studies on SAM management have largely focused on outpatient treatment costs of uncomplicated SAM, thus underestimating the overall costs of a SAM episode [7,8,9,10]. Indeed, in many of these studies, children who developed a complication during treatment were referred to inpatient care and were considered as having dropped out of the study [7,8]. Some studies have estimated the cost of inpatient care to treat complicated SAM but only included costs borne by care providers (health system or non-governmental organizations (NGOs)) [6,9] or costs associated with drugs and nutritional therapeutic foods, omitting other inpatient costs [11]. As such, a comprehensive cost analysis of SAM treatment including all relevant costs borne by all the stakeholders at the different levels of care (outpatient and inpatient) is currently lacking. In particular, the total cost of inpatient stays including the part borne by households has not been studied despite the fact that it could have a major impact on households’ livelihoods and care-seeking behaviour.

Indeed, evidence from studies on malaria and tuberculosis suggests that inpatient stays lead to a significant economic burden for households caused by an interruption of income-generating activities and additional expenses for food, transportation, and accommodation to seek treatment [12,13]. These costs are likely to be much higher than those incurred during OTPs and may require households to develop coping strategies, which may have long-lasting financial implications [14] or lead to drop-out from treatment [15,16]. In a study on malaria, disease costs were significantly associated with duration of illness, previous history of disease, and the level of the health facility and its distance from the patient’s home [17]. Factors associated with treatment costs of complicated SAM have never been explored.

Our objective was to assess the economic burden of the inpatient treatment of complicated SAM in children aged 6 to 59 months in Northern Senegal, and identify the factors associated with cost predictors. This could support advocacy for adapted strategies for SAM prevention and treatment, as well as broader health policies in Senegal and similar contexts.

## 2. Materials and Methods

We performed a retrospective cost analysis using an activity- and ingredient-based approach [18] to identify all the resources consumed during the inpatient stay of SAM patients with complications and then assigned a monetary value to them.

### 2.1. Study Context and Population

The study was carried out in Northern Senegal, in the districts of Podor and Pété in the Saint-Louis region, and in the districts of Dahra and Linguère in the Louga region (Appendix A). In this Sahelian part of Senegal, agriculture and livestock are the main livelihood. In 2022, about 4.3% and 1.8% of children aged 6 to 59 months suffered from SAM in the Louga and Saint-Louis regions, respectively [3]. Five CRENs cover the four districts included in this study, where three are located in a health centre and two are hospital-based. 

In these facilities, SAM patients are eligible for Senegal’s universal health coverage (UHC) measures for children under 5 that mainly cover: (1) consultations, vaccinations, drugs (generics only), and inpatient bed fees (for those admitted in health centres only), and (2) emergency consultations and referred cases’ consultation fees for those admitted to a hospital. Coverage is limited to XOF 4500 or USD (2020 international USD PPP) 18.7 per patient. Moreover, costs related to lab tests, non-generic drugs, and non-medical costs as well as inpatient bed fees for those admitted in hospitals are supported by households.

This study focuses on all children aged 6 to 59 months with complicated SAM hospitalized in 2020 in these five CRENs.

### 2.2. Perspective

We used a societal perspective to assess the costs borne by all stakeholders, namely the health system (supported by NGOs) and households. This perspective was chosen to provide evidence for decision-making for those most impacted by these costs [19]. 

### 2.3. Time Horizon

The time horizon was the length of the stay at the inpatient facility to treat complicated SAM.

### 2.4. Cost Valuation 

Costs were categorized into direct medical costs, direct non-medical costs and indirect costs and are presented separately for each payer (health system and households). 

#### 2.4.1. Direct Medical and Non-Medical Costs 

##### Direct Medical Costs

Direct medical costs were calculated according to Equation (1):(1)$$\begin{array}{l} Direct\;medical\;costs\\ \;\;\;\;\;\;=costs\;for\;the\;health\;system\;(including\;Personnel\;cost,\\ \;\;\;\;\;\;drugs\;cost\;covered,\;therapeutic\;foods\;cost,\\ \;\;\;\;\;\;material\;and\;consumables\;cost\;and\;hospital\;bed\;cost\;\left(for\;health\;centres\right))\\ \;\;\;\;\;\;+\;costs\;for\;households\;(including\;medical\;tests\;cost,\;drugs\;cost\;not\;covered,\\ \;\;\;\;\;\;hospital\;bed\;cost\;(for\;hospitals)) \end{array}$$


*For the health system*


Personnel costs were estimated by multiplying the average time allocated to each type of task/activity by the hourly wage of the concerned staff. 

The cost of drugs covered by the health system corresponded to the sum of the cost of generic drugs prescribed during the admission and covered by UHC measures. For each drug, we considered the cost per unit of packaging sold in pharmacies (e.g., box, vial), considering that patients had to buy a whole unit of packaging even if the entire content was not consumed.

The cost of therapeutic foods consumed as part of the nutritional treatment corresponded to the costs of F75 therapeutic milk [20], F100 therapeutic milk, and ready-to-use therapeutic food (RUTF) [21] consumed over treatment phase 1 (acute phase of restoring metabolic functions and electrolyte balance), transition phase (from phase 1 to phase 2) and phase 2 (rehabilitation phase of restoring cells to normal cellular function). These therapeutic milks/foods are given to children several times per day, with quantities proportional to their weight. The therapeutic food cost included the purchase cost, financial and operational charges, and cost related to the management of nutritional supplies by nurses at the CREN level. The latter corresponds to the personnel cost of inventorying and ensuring stock replenishment of the products once they are available at the CREN. 

Regarding material and consumables costs, the cost of single-use items (gloves, nasogastric tubes, compresses, feeding and drug administration syringes, plasters, gas strips, three-way taps and extension cords for intravenous transfusions) was calculated by multiplying the cost of each item by the frequency of use during the inpatient stay. The cost of items used for multiple patients, i.e., materials commonly used for anthropometric measures (weighing scale, height measuring system) and medical examination (thermometer, stethoscope, oximeter) was calculated based on the annual depreciation [18,22] by dividing the material’s cost by its useful lifetime obtained from expert opinion, assuming the residual value was null (formula in Appendix A). This annual depreciation value was then divided by the annual number of admissions to obtain the depreciated cost attributed to each patient. 

To estimate the cost of inpatient bed fees, we calculated the average daily bed fee by weighting bed fees by the relative number of admissions in each hospital-based CREN, and then multiplied it by the patient’s length of stay (LoS). In health centres, the health system supported a part of this cost when the UHC policy coverage limit was not reached.


*For households*


Medical test costs were calculated by adding the costs of all the tests prescribed during the inpatient stay. For each test, the average price for all CRENs was obtained by weighting the test prices by the relative number of tests in each CREN.

The costs of drugs not covered by the health system corresponded to the sum of the costs of all non-generic drugs prescribed during hospitalization and what remained of generic drug costs when the coverage limit was reached.

For inpatient bed fees, the part paid by households was the difference between the total cost and the part supported by the health system.

##### Direct Non-Medical Costs

Direct non-medical costs which are only borne by households were estimated according to Equation (2):(2)Direct non medical costs=transport cost+meals cost+hygiene kit cost

Transportation costs from the patient’s home to the CREN were calculated assuming one round trip for the accompanying person(s) and multiplying the distance between the patient’s residence and the CREN by USD 0.13 per kilometre travelled. This unit cost corresponds to the per kilometre allowance used by the general coordination group of technical and financial partners in Senegal as part of an agreement on local costs [23]. 

The costs related to the caregiver’s meals during the admission were calculated using the average daily expense for meals at each site level.

The hygiene kit cost was the cost of a basic kit including a bucket, a covered plastic mug, and some soap. These kits are often distributed free of charge at the CREN by UNICEF, but when not available, households bear the cost. 

Accommodation costs were not included as the principal accompanying caregiver occupies the same hospital bed as the SAM child and there were no data available for other caregivers. 

#### 2.4.2. Estimation of Indirect Costs 

Indirect costs correspond to the lost productivity associated with the time spent by caregivers with the child during the inpatient stay. They were estimated with the human capital approach [24] using the “proxy good” method [25]. We assumed that caregivers’ daily activities could be assimilated to housework since they are mostly housewives. We assumed there was only one accompanying person in the base case analysis and explored the presence of several accompanying persons in the sensitivity analysis.

#### 2.4.3. Total Costs

Total direct costs of an admission for complicated SAM were obtained by summing all direct cost categories for a given patient. Indirect costs were also estimated for each patient.

### 2.5. Data Collection

Demographic (age, gender, and distance between residence and CREN) and clinical (anthropometric measurements, LoS, type and mode of admission, severity of SAM at admission, reasons for admission and type of discharge from the CREN) data were retrospectively extracted from the medical records of children admitted between 1 January and 31 December 2020 in each participating CREN. Data collection was carried out between September and October 2021 using a standardized form coded in Open Data Kit application [26]. These data were used to investigate predictors of inpatient costs. 

Interviews were held with health and administrative staff to understand the repartition of costs between stakeholders.

Patients’ addresses were retrieved from their records, and the distance from home to CREN was estimated using expert opinion in order to estimate transportation costs.

Personnel and material costs were estimated by assessing, for each child, the type of care they would have received during their admission. This was based on standardized care procedures and, in the case of specific procedures, on the type of complication each child had. In order to identify times spent on each task and the materials used, individual face-to-face semi-structured interviews were conducted with each CREN’s medical staff (nurses and doctors). Medical consumables’ unit prices were retrieved from the national procurement pharmacy website [27] while the UNICEF reference prices were used for anthropometric materials [28]. As prices do not reflect the true costs for CRENs as it does not include financial (clearance taxes, insurance and custom) and operational (handling, processing and storage) costs, we estimated the cost for CRENs by adding a 50% financial and operational charge to the purchased price based on expert opinion. This percentage was extensively varied in sensitivity analyses. 

The costs of drugs, medical tests, and therapeutic foods were based on drugs prescribed, tests carried out, and nutritional therapeutic foods consumed during hospitalization and notified in medical records. Data on drug prices were obtained from the national procurement pharmacy website [27] and from local private pharmacies. Administrative staff were interviewed to collect data on medical test pricing. Therapeutic food prices were UNICEF reference prices [28] plus financial and operational charges as previously described.

Nurses, who are closest to patients and therefore have a better knowledge of actual practices, were asked to provide estimates on meals’ cost and hygiene expenses usually borne by caregivers during the stay.

For indirect costs, we used an estimate of the monthly income for domestic helpers, retrieved from the harmonized survey on household living conditions in Senegal [29], i.e., USD 170 [2020 US international dollar using purchasing power parity (PPP) exchange rate] per month for full-time work.

Costs were mostly collected in XOF and converted into 2020 US international dollars, using a PPP conversion rate of USD 1 for XOF 240.317 [30]. Costs initially expressed in euros were first converted into XOF based on a fixed rate of EUR 1 for XOF 655.957 and then into international dollars.

### 2.6. Analysis

Individual clinical and demographic data were managed and analysed with STATA software version 16 [31]. Data retrieved during interviews were analysed using Microsoft Excel 2016. The sensitivity analysis was carried out on TreeAgePro software version 2021 R1.1 [32]. 

#### 2.6.1. Descriptive Analysis

Patients’ demographic and clinical characteristics were described using numbers and percentages for categorical variables and means and standard deviations (SDs), and median and interquartile range for continuous variables. Costs were described using means and SDs. Subgroup analyses were carried out by facility status (health centre vs. hospital). 

#### 2.6.2. Sensitivity Analysis

We performed a univariate deterministic sensitivity analysis as well as a scenario analysis to assess the uncertainty surrounding our cost estimation. For the univariate sensitivity analysis, we performed a tornado analysis by individually varying the study parameters across a range of plausible values (values and assumptions in Table 1) to identify the main cost drivers. 

The scenario analysis consisted of a best- and worst-case scenario analysis where we simultaneously varied all parameters using the minimum values for the best-case scenario and the maximum values for the worst-case scenario. 

#### 2.6.3. Factors Associated with Cost 

We carried out a multivariable generalized linear model with gamma distribution and a log link function to investigate how patients’ demographic and clinical characteristics influenced direct (medical and non-medical) costs. This type of model was chosen to handle cost data that were right-skewed and non-negative [34]. The covariates of interest were gender, age, status of the facility hosting the CREN, SAM severity, inpatient mortality, number of complications, and complications among the most frequent (diarrhoea, dehydration, acute respiratory infection, and anaemia) considered individually as dummy variables. We first carried out a univariate model, and the variables found to be significantly associated with total direct cost at a significance level of 20% were introduced in the multivariable model.

### 2.7. Ethical Considerations

This study was approved by the Senegalese National Ethical Committee for Health Research (CNERS). Approval to access archival documents (medical records) was obtained from districts’ authorities, and precautions to ensure anonymous access to these records were observed throughout the process. Patients’ identity and caregivers’ contact information were not collected. 

## 3. Results

### 3.1. Characteristics of the Study Population

A total of 140 children aged 6 to 59 months were hospitalized in the five CRENs during the study period. There were more boys (53.6%) and the mean age was 18 months (SD = 9.4) (Table 2). Over two-thirds of children lived more than 10 kilometres away from the CREN, with over a quarter living over 50 km away. The majority of cases were new admissions (97.9% versus 2.1% of relapses) and 84.3% of children were considered successfully treated when discharged from the CREN (90.2% and 80.9% in health centres and hospitals, respectively). Nearly 8% of children dropped out of CRENs, and the drop-out rate was higher in hospital-based CRENs (10.1% vs. 3.9% health-centre-based CRENs) (Appendix A). Mean LoS was 5.3 days (SD = 3.2). The most common complications that justified inpatient care were diarrhoea (55.7%), dehydration (29.3%), co-occurrence of diarrhoea and dehydration (20.7%), and anaemia (22.1%). Approximately 45.0% of children suffered from more than two complications. 

### 3.2. Base Case Results of Costs Associated with Inpatient Stay

The cost units used for this analysis are presented in Table 1.

Mean total cost per child of the CREN inpatient stay was USD 431.9 (SD = 204) (2020 current international USD PPP) (Table 3). This mostly corresponded to direct costs (92%), which amounted to USD 397.2, with the biggest part being direct medical costs (81% of direct costs). Direct medical costs were mainly borne by the health system (63%) and the rest was paid by households. Households also bore the entirety of direct non-medical costs, which represented 19% of direct costs. This corresponded to a total out-of-pocket expense of USD 195.6 (SD = 103.7), representing 45.3% of the total cost. 

The biggest cost component was that of personnel (33%) and it was similar across CRENs. Two major cost components were borne by households, namely lab tests (11.4% of total cost) and meals for the accompanying caregiver (11.2%) (Table 3).

In addition to these costs, caregivers faced productivity losses equivalent to USD 34.7 (SD = 21.0), representing 8% of total cost.

When considering the status of the facility hosting the CREN, total costs were higher in hospital-based CRENs (USD 473.4 (SD = 45.2)) compared to health-centre-based CRENs (USD 357.6 (SD = 47.9)), with a marked difference for direct medical costs and more importantly for the proportion borne by households (Table 3). In hospitals, households’ out-of-pocket expenses for direct medical costs were more than four times higher than those borne in health centres (USD 37.0 (SD = −3.5) vs. USD 165.9 (SD = 15.3)). 

### 3.3. Sensitivity Analysis

The tornado analysis (Figure 1) showed that the parameters with the greatest impact on costs were the number of accompanying caregivers during the stay, the average LoS in phase 1 treatment, and the hourly wage of the nurses (which depends on seniority).

When applying best- and worst-case scenarios to the study parameters, mean total costs associated with the inpatient stay for complicated SAM ranged from USD 260.6 to USD 761.2 (Appendix A).

### 3.4. Factors Associated with Cost

In the univariate analysis, gender, inpatient mortality, status of the facility hosting the CREN, diarrhoea, dehydration, acute respiratory infection, and anaemia were significantly associated with total direct costs (*p*-value < 0.2). In the multivariable analysis, the association remained significant for gender, diarrhoea, anaemia, inpatient death, and status of the facility hosting the CREN (Table 4). Direct costs were significantly lower by 20.3% for boys compared to girls. Diarrhoea significantly increased direct costs by 27%. Likewise, when children were treated for anaemia, direct costs significantly increased by USD 196.2 (SE = 61.4) or 49.4%. Treatment in a hospital-based CREN (versus a health centre) increased direct costs by 26% or USD 103.2 (SE = 31.5).

## 4. Discussion

This study aimed to estimate the average cost associated with a CREN inpatient stay to treat complicated SAM in children aged 6 to 59 months in Northern Senegal. The mean societal cost per patient was USD 431.9 (SD = 203.9) for a mean LoS of 5.3 days (SD = 3.2).

Previous studies estimated the costs of SAM inpatient care mostly by using a provider perspective and therefore omitted direct medical and non-medical costs borne by households as well as indirect costs. A provider perspective was followed by Isanaka et al. (2017) [6] in Niger, which uses a treatment protocol similar to Senegal’s but with a stand-alone program providing care only to SAM children with a dedicated NGO staff. It found a hospital cost of EUR 134.5 or USD 367.3 (2020 US international USD PPP) for a mean LoS of 4.6 days. This cost is higher than ours, with only USD 194.8 borne by the health system. This difference is likely due to higher salaries paid to NGO personnel and the inclusion of infrastructure and logistic support costs for a stand-alone SAM inpatient facility.

In our study, the largest expense was personnel costs (33% of total cost). This is because children with complicated SAM require close surveillance, time to prepare the nutritional food, supervised feeding (up to eight meals per day), and time to fill out follow-up forms. Indeed, in our study, nurses spent on average 7.2 h with each child during the entire stay, equivalent to one workday or 18% of weekly working time. In the absence of dedicated staff for complicated SAM children, admission of simultaneous cases may lead to a heavy workload and impact quality of care. Comparing personnel cost across different contexts is a delicate process as it depends on wages’ level (which varies according to the type of staff involved in care provision), their status (public sector or NGOs), their seniority, and time devoted to the care activities/tasks. The latter may depend on treatment protocols, the child’s clinical condition and the CREN’s organization (isolated CREN with a dedicated team or CREN integrated into a paediatric service).

Although cost categories vary between studies, they all include the costs of therapeutic food consumed during the inpatient stay. These costs were similar to what we found: USD 39.1 (2020 US USD PPP) in Niger [6] and USD 47.0 (2020 US USD PPP) in Burkina Faso [11], compared to USD 35.5 in our study.

By including households’ perspective, our study highlights the hidden financial burden of SAM in a context where the treatment is said to be “free of charge for families”. Although therapeutic foods are free, hospitalized children also require medical treatment paid by households, in addition to the direct non-medical and indirect costs they bear. Households’ out-of-pocket expenses represented 45.3% of total cost and amounted to USD 195.6 (SD = 103.6), representing 13% of the annual consumption expenditure (food and non-food expenses) for households living in rural areas [USD 1540 (2020 US USD PPP) in 2021] [29]. This burden is consequent, given the high mortality rate during the stay (7.1%) and upcoming ambulatory care costs following the CREN discharge.

Although our population was eligible for UHC measures for children under five in Senegal [35], which cover some components with a coverage limit per patient, the remaining amount of out-of-pocket expenses borne by households raises concerns about the ability of the current UHC measures to effectively mitigate the disease burden for households with hospitalized children. When patients continue to face high expenses in contexts of free-user-fee policies, this is often due to implementation gaps [36,37]. These gaps include generic drug supply shortages from the health facility pharmacies, forcing patients to buy drugs from private pharmacies, and healthcare staff failing to adequately apply exemption policies [36]. The supply of drugs may have been a concern in our study’s sites. Indeed, in the review of medical records, we found that the majority of drugs consumed were not generic drugs and therefore not covered by the exemption when the facility was eligible (health centres). Higher costs are borne in hospitals, where UHC measures do not cover drugs, lab tests, and hospital bed fees. Even in health centres, direct medical costs remained high as lab tests and non-generic drugs were also not covered. To effectively reduce the financial burden borne by households, there is a need to generalize UHC measures to all facilities regardless of their status, and review what is covered by the fee exemptions.

An exploration of favourable and unfavourable scenarios showed that the total mean cost of an inpatient stay at a CREN could range from USD 260.6 to USD 761.2. The sensitivity analysis showed that households’ costs were mostly driven by the number of people staying with the child. This not only implies more expenses for meals and loss of productivity, but also has implications in terms of household reorganization regarding domestic workloads during the hospitalization. In addition, treatment cost may lead to long-term financial consequences for households and increase families’ nutritional insecurity and the risk of relapse, leading to additional burden for the impacted families. A qualitative study would provide a better understanding of the financial but also social consequences for a family with a hospitalized child. For those families, strong protective policies like a health allowance and a total fee exemption for SAM treatment should be implemented to avoid further financial insecurity and the vicious circle of relapse.

We found that the total direct cost of a CREN stay was significantly associated with gender, status of the facility hosting the CREN, the status of being treated for diarrhoea and anaemia, and inpatient death.

Being a female patient significantly increased direct costs. Girls with SAM may experience more severe complications, potentially due to biological differences or unequal access to healthcare. More investigation is needed to explore these potential reasons and construct targeted interventions to address them.

A stay in a hospital-based CREN led to a USD 103.2 (SD = 31.5) increase in direct (medical and non-medical) costs compared to a stay in a CREN located in a health centre. Similar results were found with regard to malaria treatment in Ethiopia [17]. In our study, mean direct medical costs between health centres and hospitals were different, specifically regarding lab tests and drug costs. For lab tests, the cost difference may be due to a greater technical capacity, which could have facilitated more investigations. If these investigations led to additional diagnoses, this could lead to higher drug consumption and drug costs. Additionally, complications treated at hospital-based CRENs may have been more severe (and requiring more drugs) as mortality was higher in hospital-based CRENs (9.0%) compared to health-centre-based CRENs (3.9%). Nonetheless, the higher costs incurred in hospitals could have contributed to the higher drop-out rate observed in hospital-based CRENs (10.1% vs. 3.9% health-centre-based CRENs).

The cost associated with a CREN stay was not significantly associated with the severity of SAM, but with the type of complications. This is not surprising as CREN admission does not depend on SAM severity but on the presence of complications. Thus, a child may be discharged from the CREN as an uncomplicated SAM case to continue treatment in an OTP. Regarding complications, diarrhoea and anaemia were significantly associated with an increase in direct costs. As anaemia is a common complication in SAM children requiring inpatient care Thakur et al. (2014) [38], specific prevention measures like iron-rich diet promotion should be undertaken early during SAM OTP.

In our sample, the most common complications that justified CREN admission were diarrhoea (55.7%), dehydration (29.3%), and the co-occurrence of diarrhoea and dehydration (20.7%). This is consistent with a Senegalese study by Seck et al. (2021) [39] on SAM management in the CREN of the Diamniadio paediatric hospital close to Dakar that reported a prevalence of diarrhoea of 57.5% but a higher prevalence of dehydration (40.3%). Having diarrhoea also significantly increased the direct cost. This high burden of diarrhoea among hospitalized SAM cases raises the necessity to invest in diarrhoea prevention when children are admitted in OTPs and pursue their treatment in the less controlled household setting with a higher risk of infections from unclean water and other environmental hazards [40]. Investing in a WASH and Nutrition strategy [41] that includes the provision of a package of water treatment and hygiene promotion at the admission of SAM children in the OTPs could help reduce referrals due to diarrhoea [42] and consequently avoid inpatient costs for the health system and affected households.

To our knowledge, this study is the first to investigate the costs of managing SAM with complications from a societal perspective and to study the associated factors and main cost drivers. Our results will facilitate the estimation of the full financial burden of a SAM episode and the cost-effectiveness of potential prevention measures. An extensive activity and ingredient costing approach coupled with clinical and demographic characteristics allowed us to identify clear drivers of cost and propose tailored mitigation strategies. Furthermore, results will support advocacy efforts of financial protection for families and for more resources allocated to SAM treatment.

However, this study has some limitations. First, personnel cost, which was the main cost driver, is subject to some variability due to factors such as staff type, employment status, and seniority. Although this has been considered in sensibility analyses by applying different levels of wages, comparisons with other studies must be performed with caution. Secondly, some costs could not be included due to data unavailability. These include capital costs of buildings and administrative equipment, as well as other expenses incurred to keep the CREN in operation (cleaning costs, equipment for children’s playrooms). In addition, intangible costs (like anxiety, stress, and school drop-out) were not included in this study. Furthermore, assumptions about households’ lost wages could have affected the estimation of the costs borne by households. Further investigations are currently ongoing to gather more precise data on these aspects and capture the complete financial burden for households affected by SAM. Finally, our sample size may have affected the precision of the model on the factors associated with direct costs. Although the sample covered five SAM inpatient facilities spread across the north of Senegal, it is not possible to state whether those costs are representative of all of Senegal. Further research involving a broader sample of facilities across different geographical regions within Senegal would provide a clearer picture of the national economic burden of severe acute malnutrition with complications and its regional variations. Moreover, longitudinal studies to track the long-term economic impact of SAM treatment on households could provide a more comprehensive economic burden of SAM for affected communities.

## 5. Conclusions

Our study highlights the burden of treating complicated SAM for the health system and households in Northern Senegal. The mean societal cost per patient was USD 431.9 (2020 USD PPP), with 45.3% supported by households as out-of-pocket expenses. Adequate prevention strategies must be put in place to target the avoidable complications of SAM in order to avoid hospital referrals and their high associated costs. Larger exemption fees and tailored social assistance schemes should be established to help SAM treatment adherence, and protect families from the disease’s high financial burden. There is a critical need for integrating economic evaluations in the planning and implementation of health and nutrition interventions in order to develop more effective and sustainable strategies that not only address the medical needs of the population but also alleviate the economic strain on families and the health system.

## Figures and Tables

**Figure 1 nutrients-16-02192-f001:**
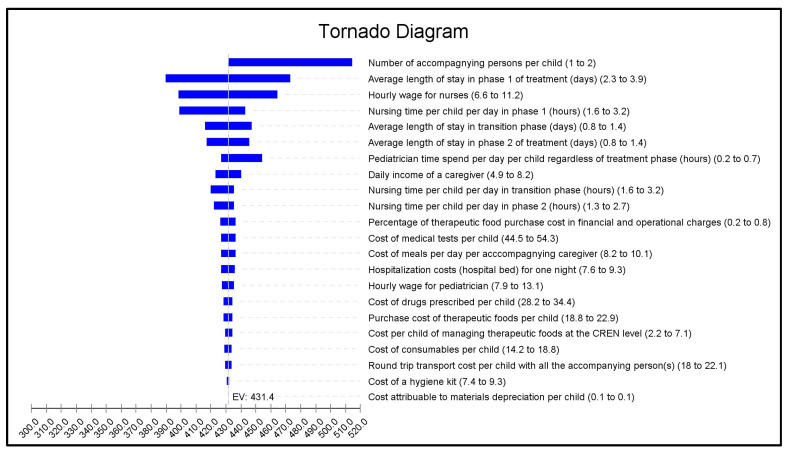
Univariate sensitivity analysis, tornado diagram of the mean cost of inpatient stay to treat complicated severe acute malnutrition. For each variable, the first value in brackets represents the base case value, the second is the minimum value, and the third is the maximum value. The impact of each variable on cost is represented by the width of the horizontal bar. The extremities of each bar indicate the maximum and minimum costs corresponding to the application of the worst and best case scenarios for each variable. The dotted line represents mean cost in the base case analysis.

**Table 1 nutrients-16-02192-t001:** Unit costs and quantities used for cost estimation, base case values, assumptions, ranges used in the sensitivity analyses and data sources.

Parameter Description	Base Case Value [Best-Case Scenario Value—Worst-Case Scenario Value]	Assumption for Min Values (Best-Case Scenario)	Assumption for Max Values (Worst-Case Scenario)	Source
LoS1: Average length of stay at phase 1 ^†^ of treatment (days)	3.1 [2.3–3.9]	We applied −25% to the base case value	We applied +25% to the base case value	- Medical records - Protocol of management of SAM in Senegal
LoS2: Average length of stay at transition phase ^‡^ (days)	1.1 [0.8–1.4]			
LoS3: Average length of stay at phase 2 ^§^ of treatment (days)	1.1 [0.8–1.4]			
a1: Purchase cost of therapeutic foods per child	USD 20.8 [18.3–22.9]	We applied −10% to the base case value	We applied +10% to the base case value	UNICEF reference price for each type of therapeutic food, retrieved from UNICEF’s supply website (consulted on March 2022) [28]
a2: Cost of managing therapeutic foods at the CREN level per child This corresponds to the time dedicated to nutritional products’ management activities (inventorying and ensuring stock replenishment) performed by nurses at the CREN level, converted into costs	USD 4.2 [2.1–7.1]	We used the 10th percentile of the values declared by the nurses for the time spent on therapeutic foods’ management activities; the hourly wage considered was the best-case value for nurses’ hourly wage	We used the 90th percentile of the values declared by the nurses for the time spent on therapeutic foods’ management activities; the hourly wage considered was the worst-case value for nurses’ hourly wage	- Time allocation data collected during interviews with nurses - Hourly wage from votresalaire.org (consulted on May 2022) [33]; the salary considered is the median value of the salary range for a nurse with 5 years of seniority
p: Percentage of therapeutic food purchase cost added to cover financial and operational charges These charges include freight costs, clearance taxes, insurance, customs, handling, processing, and storage costs	50% [25–75%]	We applied −25% to the base case value	We applied +25% to the base case value	Assumption based on previous study [10] and expert opinion
A = a1 × (1 + p) + a2: Total cost (purchase and associated charges) of therapeutic food per child		NA	NA	NA
B: Cost of single-use consumables per childWe summed the costs of all single-use items consumed by each child according to their clinical characteristics and length of stay	USD 16.4 [14.2–18.8]	The cost of each consumable was reduced by 10%; the length of stay considered was that of the best-case value	The cost of each consumable was increased by 10%; the length of stay considered was that of the worst-case value	Identification of the consumables, their frequency of use, and when they are used, which is assessed during interviews with the medical staff; consumables’ prices from the national procurement pharmacy [27]
C: Cost of drugs prescribed per child Corresponds to the average costs of drugs prescribed during hospitalization, including the part borne by the health system (USD 4.6) by households (USD 26.7)	USD 31.3 [27.9–34.7]	We applied −10% to the base case cost values	We applied +10% to the base case cost values	Drug prices retrieved from the national procurement pharmacy website [27] and from local private pharmacies
D: Cost of medical tests per child Corresponds to the average costs of medical tests prescribed to all children in the sample	USD 49.2 [44.5–54.3]	We applied −10% to the base case cost values	We applied +10% to the base case cost values	Interviews with the CREN medical and administrative staff
e1: Hospitalization costs (hospital bed) for one night Corresponds to the average daily bed fee obtained by weighting bed fees by the relative number of admissions in each CREN	USD 8.4 [7.6–9.3]	We applied −10% to the base case cost values	We applied +10% to the base case cost values	Interviews with CRENs’ medical and administrative staff
E = e1 × (LOS1 + LOS2 + LOS3): Hospitalization cost (payment for hospital bed occupation) per child		NA	NA	NA
F: Cost of multiple-use materials per child Corresponds to the sum of multiple-use materials’ depreciated values attributable to one child	USD 0.08 [0.08–0.10]	Estimated with: - Materials’ prices reduced by 10%; - The percentage of the purchase cost as financial and operational charges reduced by 25%; - The lowest value declared by the experts for the useful time of each of the materials	Estimated with: - Materials’ prices increased by 10%; - The percentage of the purchase cost as financial and operational charges increased by 25%; - The highest value declared by the experts for the useful time of each of the materialsThe cost of each consumable was increased by 10%	Material prices retrieved from UNICEF’s supply website (consulted on March 2022) [28] Materials’ useful time based on expert opinion Percentage of purchase cost added to cover financial and operational charges based on expert opinion and previous study.
g1: Nursing time per child per day in phase 1 (hours)	2.8 [1.6–3.2]	We used the sum of the 10th percentile values of nurses’ declarations regarding the time allocated to each type of activity per child per day	We used the sum of the 90th percentile values of nurses’ declarations regarding the time allocated to each type of activity per child per day	Interviews with CRENs’ nurses
g2: Nursing time per child per day in transition phase (hours)	2.3 [1.3–2.7]			
g3: Nursing time per child per day in phase 2 (hours)	2.3 [1.3–2.7]			
g4: Paediatrician time spent per child per day regardless of treatment phase (hours)	0.3 [0.2–0.4]	We used the 10th percentile value of paediatricians’ declarations regarding the time spent per child per day	We used the 90th percentile value of paediatricians’ declarations regarding the time spent per child per day	Interviews with CRENs’ paediatricians
w1: Hourly wage for nurses Corresponds to the median value of the salary range for a nurse with 5 years of seniority	USD 8.8 [6.5–11.2]	We applied −25% to the base case value	We applied +25% to the base case value	Hourly wage from votresalaire.org (consulted on May 2022) [33]
w2: Hourly wage for paediatricians Corresponds to the median value of the salary range for a paediatrician with 5 years of seniority	USD 10.5 [7.9–13.1]	We applied −25% to the base case value	We applied +25% to the base case value	Hourly wage from votresalaire.org (consulted on May 2022) [33]
G = (w1 × (LOS1 × g1 + LOS2 × g2 + LOS3 × g3) + w2 × (LOS1 + LOS2 + LOS3) × g4): Personnel cost per child		NA	NA	NA
Direct medical costs per child (DMC = A + B + C + D + E + F + G)		NA	NA	NA
h1: Cost of meals per day per accompanying caregiver Corresponds to the average cost of meals sold around the CRENs, obtained by weighting each cost by the relative number of admissions in each CREN	USD 9.1 [8.2–10.1]	We applied −10% to the base case value	We applied +10% to the base case value	Interviews with CRENs’ nurses
h2: Number of accompanying persons per child	1 [1–2]	As in the base case analysis, we assumed there was one accompanying caregiver per child	We assumed there were two accompanying caregivers per child	Interviews with CRENs’ nurses and assumption
H = h1 × h2 × (LOS1 + LOS2 + LOS3): Average meals’ cost for the accompanying caregiver(s) throughout entire stay		NA	NA	NA
I: Round-trip transportation cost per child with the accompanying person(s)	USD 20.0 [18.0–22.1]	We applied −10% to the base case value	We applied +10% to the base case value	The distance from the patient’s home to the CREN was estimated using their address present in their medical records The unit cost per kilometre travelled came from the agreement on the standardization of local costs applied to local staff, used by the general coordination group of technical and financial partners in Senegal [27]. As this is used for employees, we requested an expert opinion to confirm the plausibility of the use of the same amount for caregivers’ transportation cost estimation
J: Cost of a hygiene kit	USD 8.5 [7.4–9.3]	We applied −10% to the base case cost value	We applied +10% to the base case cost value	Interview with CRENs’ nurses
DNMC = H + I + J: Direct non-medical costs per child		NA	NA	NA
k1: Daily income of a caregiver	USD 6.6 [4.9–8.2]	We applied −25% to the base case value	We applied +25% to the base case value	Estimated with the monthly income corresponding to the caregiver’s status, retrieved from the harmonized survey on household living conditions in Senegal [17]
k2 = k1 × (LOS1 + LOS2 + LOS3: Loss of productivity for a caregiver		NA	NA	NA
IC = k2 × h2: Indirect costs (loss of productivity for the accompanying person(s))		NA	NA	NA

NB: All costs are expressed in 2020 US international dollars, using the purchasing power parity (PPP) conversion rate of XOF vs. dollar: USD 1 = XOF 240.317. ^†^ Phase 1 corresponds to the acute phase of treatment to initiate recovery of metabolic functions and restore electrolyte nutritional balance. F75 therapeutic food is used at this phase. ^‡^ Transition phase: transition from phase 1 to phase 2. ^§^ Phase 2 corresponds to the rehabilitation phase consisting of restoring cells to normal cellular function with the high-protein F100 therapeutic milk and ready-to-use therapeutic food (RUTF). Abbreviation: NA, not applicable; SAM, severe acute malnutrition.

**Table 2 nutrients-16-02192-t002:** Characteristics of children aged 6 to 59 months admitted for complicated severe acute malnutrition in Northern Senegal between January and December 2020 (N = 140).

**Demographic characteristics**		
Gender, n (%)		
Male	75 (53.6%)	
Female	65 (46.4%)	
Age in months, mean (SD), median (IQR)	18.1 (9.4)	17 [11.5–24]
Residence distance from the CREN, n (%)		
≤10 km	43 (30.7%)	
11–50 km	65 (46.4%)	
>50 km	32 (22.9%)	
**Anthropometric measurements**		
Weight at admission, in kg, mean (SD), median (interquartile range)	6.9 (1.8)	6.7 [5.6–7.8]
Height at admission, in cm, mean (SD), median (interquartile range)	75.1 (9.2)	75 [70–80]
Severe acute malnutrition severity, n (%)		
Severely wasted (−4 < Z-score ≤ −3)	44 (31.4%)	
Very severely wasted (Z-score ≤ −4)	96 (68.6%)	
**Characteristics of the admission**		
Admission type, n (%)		
New admission ^†^	137 (97.9%)	
Relapse ^‡^	3 (2.1%)	
Admission mode, n (%)		
Referral from UREN ^§^	117 (83.6%)	
Internal referral ^¶^	23 (16.4%)	
Type of discharge from the CREN, n (%)		
Successfully treated ^#^	118 (84.3%)	
Dropped out ^||^	11 (7.9%)	
Medical referral ^††^	1 (0.7%)	
Dead	10 (7.1%)	
Length of stay, in days, mean (SD), median (interquartile range)	5.3 (3.2)	5 [3–7]
**Complications motivating the admission (not mutually exclusive), n (%)**		
Diarrhoea	78 (55.7%)	
Dehydration	41 (29.3%)	
Acute respiratory infection	27 (19.3%)	
Anaemia	31 (22.1%)	
Anorexia	14 (10.0%)	
Oedema	5 (3.6%)	
Other ^‡‡^	12 (8.6%)	
Number of complications, n (%)		
1	77 (55.0%)	
2	47 (33.6%)	
3–4	16 (11.4%)	
Number of complications, mean (SD), median (interquartile range)	1.6 (0.7)	1 [1–3]

^†^ Children admitted to the CREN for the first time following an episode of complicated severe acute malnutrition. ^‡^ Children admitted for a second complicated severe acute malnutrition episode occurring within three months of recovery from a previous episode treated at any inpatient or outpatient facility. ^§^ Children referred by a unit of rehabilitation and nutritional education (UREN), i.e., severe acute malnutrition outpatient treatment facilities. ^¶^ Child diagnosed during a routine consultation at the health centre or hospital hosting the CREN. ^#^ Complication(s) treated and child referred back to UREN to continue severe acute malnutrition outpatient treatment. ^||^ Decision to leave the CREN against medical advice ^††^ Referral to another inpatient facility. ^‡‡^ Other complications include hyperthermia, oral candidiasis, sickle cell disease, psychomotor disability, trisomy, and asthma. Abbreviations: CREN = Centre for Rehabilitation and Nutritional Education (inpatient treatment centre for SAM in Senegal). SD = Standard deviation. UREN = Unit for Rehabilitation and Nutritional Education (outpatient treatment centre for SAM in Senegal). km = kilometre. IQR: Interquartile range.

**Table 3 nutrients-16-02192-t003:** Mean cost of inpatient stay to treat complicated severe acute malnutrition in children aged 6 to 59 months in Northern Senegal in 2020 by type of care and category of payers.

	Mean Cost (SD) (in 2020 US International USD ^†^)	% of Total Cost	Mean Cost for Health-Centre-Based CREN	Mean Cost for Hospital-Based CREN
**Direct medical costs**	**320.5 (154.3)**	**74.2%**	**249.1 (124.7)**	**360.5 (155.5)**
**Borne by the health system**	**201.7 (103.7)**	**46.7%**	**212.1 (35.3)**	**194.6 (16.3)**
Personnel	142.7 (84.4)	33%	141.2 (87.5)	142.7 (83.2)
Drugs	4.6 (6.8)	1.1%	12.6 (5.0)	0.0 (0.0)
Therapeutic foods	35.5 (21.9)	8.2%	35.4 (20.3)	35.5 (22.9)
Hospital bed	2.2 (4.2)	0.5%	6.1 (5.0)	0.0 (0.0)
Material and consumables	16.5 (4.0)	3.8%	16.8 (4.3)	16.4 (3.9)
**Borne by households**	**118.9 (82.5)**	**27.5%**	**37.1 (3.5)**	**165.9 (15.3)**
Drugs	26.7 (24.5)	6.2%	3.2 (5.9)	40.1 (20.7)
Medical tests	49.3 (34.6)	11.4%	18.0 (22.7)	67.2 (26.4)
Hospital bed	42.9 (36.7)	9.9%	15.8 (14.3)	58.5 (38.0)
**Direct non-medical costs**	**76.7 (35.7)**	**17.8%**	74.0 (32.8)	78.3 (37.7)
**Borne by households**	**76.7 (35.7)**	17.8%	74.0 (32.8)	78.3 (37.7)
Caregiver(s) meals	48.4 (29.3)	11.2%	17.5 (15.8)	21.5 (18.2)
Transportation	20.0 (17.4)	4.6%	48.2 (30.2)	48.5 (28.9)
Hygiene kit	8.3 (0.0)	1.9%	8.3 (0.0)	8.3 (0.0)
**Total direct costs**	**397.2 (30.0)**	**92.0%**	**323.1 (152.5)**	**438.8 (188.8)**
**Indirect costs (caregiver productivity loss)**	**34.7 (21.0)**	**8.0%**	**34.5 (21.6)**	**34.7 (20.7)**
**Total costs**	**431.9 (203.9)**	**100%**	**357.6 (47.9)**	**473.4 (45.3)**
**Borne by the health system**	**201.6 (103.7)**	**46.7%**	**212.1 (35.3)**	**194.6 (16.3)**
**Borne by households (out-of-pocket)**	**195.6 (103.6)**	**45.3%**	**111.1 (49.6)**	**244.1 (95.1)**
**Caregiver(s) productivity loss**	**34.7 (21.0)**	**8.0%**	**34.5 (21.6)**	**34.7 (20.7)**

^†^ 2020 US international dollar using purchasing power parity (PPP) exchange rate of XOF vs. US dollar: USD 1 = XOF 240.317. Abbreviations: CREN = Centre for Rehabilitation and Nutritional Education (SAM inpatient treatment centre in Senegal). SD = Standard deviation. US = United States.

**Table 4 nutrients-16-02192-t004:** Factors associated with total direct costs for an inpatient stay to treat complicated severe acute malnutrition in children aged 6 to 59 months in Northern Senegal.

	Coefficient	Standard Error	*p*-Value
Gender (reference: female)			
Male	−80.6	31.3	0.010 *
Age (in months)	−0.1	1.6	0.960
SAM severity (reference: WFH ≤ 3 Z-score)			
Very severely malnourished (≤−4 Z-score)	−49.1	33.5	0.143
Diarrhoea (reference: no)	107.2	35.8	0.003 *
Dehydration (reference: no)	83.2	50.2	0.097
Acute respiratory infection (reference: no)	10.0	46.1	0.828
Anaemia (reference: no)	196.2	61.4	0.001 *
Number of complications (reference: 1 complication)			
2 complications	−65.0	47.7	0.173
≥3 complications	−105.5	65.6	0.108
Inpatient death (reference: no)	−178.3	36.5	<0.001 *
Status of the facility hosting the CREN (reference: health centre)			
Hospital	103.2	31.5	<0.001 *

NB: Costs are expressed in 2020 US international dollars, using the purchasing power parity (PPP) conversion rate of XOF vs. dollar: USD 1 = XOF 240.317. * significant at a 5% level. Abbreviations: CREN = Centre for Rehabilitation and Nutritional Education. WFH: Weight for height.

## Data Availability

The data presented in this study are available upon request from the corresponding author. The data are not publicly available due to ethical reasons.

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
