# Peer review of "The Economic Burden of Severe Acute Malnutrition with Complications: A Cost Analysis for Inpatient Children Aged 6 to 59 Months in Northern Senegal"

_nutrients, 2024, doi:10.3390/nu16142192_

Round 1

Reviewer 1 Report

Comments and Suggestions for Authors

In the manuscript submitted to me for review entitled "The economic burden of severe acute malnutrition with complications: a cost analysis for inpatient children aged 6 to 59 months in Northern Senegal the authors Bibata Wassonguema, Dieynaba S N’Diaye, Morgane Michel, Laure Ngabirano, Severine Frison, Matar BA, Françoise Siroma, Antonio V Brizuela, Martine Audibert and Karine Chevreul present an estimate of the costs of inpatient treatment of children with complications, aged 6 to 59 months, in Northern Senegal as a result of severe acute malnutrition (SAM).

The study included 140 children hospitalized from January to December 2020. The study was approved by the Senegalese National Ethics for Health Research (CNERS). Permission to access archival documents was also obtained from the regional authorities. In addition to addressing the problem of child mortality due to malnutrition in Senegal, which cannot be solved at the moment, the study shows how much financial resources are currently allocated in an effort to prevent at least some of the child mortality. It is also indicated the need for households to also be involved in covering part of the costs, because the provided finances are insufficient. This and similar studies could attract the attention of relevant national authorities to plan new approaches to combating child mortality, such as considering the possibility of at least part of the intended financial means being planned as social benefits for families with children, thereby at least to a certain extent to avoid child malnutrition, instead of, for example, money being allocated to caregivers for children who are already in a critical condition.

The collected data are presented using 1 figure and 4 tables. The conclusions drawn by the authors are well-formed and an accurate summary of the information presented. To support their research, the authors used 40 references that present information from studies published mostly in the last two decades. About 1/2 of the total references are from the last 5 years, indicating that the topic under review is relatively new and current and would be of interest to Nutrients readers. I did not notice any redundant self-citations, all references used are appropriate and necessary for the preparation of the manuscript.

My remarks and recommendations to the authors are:

1. In Table 1, the symbol "*" is indicated in many places, but it is not given below the table what it means. At the same time, there are three other symbols explained below the table that I did not see anywhere in the table. Let the authors look at all the symbols made and see where they are in their proper place.

2. I am somewhat concerned about the fact that consent was not obtained from the individuals included in the study. It is true that the study is related to the financial part of the problem under consideration, but it still goes back to real 140 children.

3. Figure 1 is perhaps represented 2 times.

4. There is a link in the Supplementary Materials section, but it is not active for me personally. Why is the information not presented in a separate Supplementary file?

Author Response

Authors: Thank you for reviewing our manuscript. We appreciate the positive and constructive feedback. We have provided a point-by-point response for your consideration. Please see the attachment.

Reviewer 2 Report

Comments and Suggestions for Authors

Dear Authors, please find below my detailed feedback aimed at refining and strengthening your manuscript for its next revision.

Abstract

  1. Clarity and Precision: The abstract must clearly outline the study's objectives, methodology, significant results, and implications. Currently, it lacks precision in detailing the study’s methodological approach and specific findings. To improve, specify the model used and highlight key numerical findings directly in the abstract.
  2. Keyword Relevance: Ensure that the keywords accurately reflect the manuscript’s content and focus. Consider adding more specific terms related to the financial implications and geographic specifics (e.g., "health economics", "Northern Senegal").
  3. Objective Alignment: The objectives mentioned in the abstract and introduction should align perfectly. Currently, the abstract suggests a broader focus than is detailed in the introduction.
  4. Impact Statement: Include a statement of impact that succinctly describes the importance and potential implications of the findings for health policy in Senegal and similar contexts.

Introduction

  1. Context and Significance: Begin by highlighting the global and regional burden of severe acute malnutrition (SAM), emphasizing its relevance in Senegal. Mention statistics to frame the issue within the broader context of public health challenges in West Africa.
  2. Research Gap: Clearly articulate the gaps in the existing literature, specifically the lack of comprehensive cost analyses that incorporate both direct and indirect costs from a societal perspective. Stress the importance of understanding these costs in the context of Northern Senegal.
  3. Study Objectives: Define the primary objectives of your study, which are to estimate the economic burden of SAM with complications in Northern Senegal and to identify the main predictors of cost variability.
  4. Relevance to Policy and Practice: Conclude the introduction by linking the research objectives to potential policy implications, suggesting that this study could inform more effective health policy strategies in Senegal and similar settings.

Methods

  1. Study Design: Describe the design of your study, mentioning that it is a retrospective cost analysis. Detail the inclusion criteria, such as age range and specific health conditions (e.g., SAM with complications).
  2. Data Sources and Collection: Explain the sources of your data, including hospital records and interviews with healthcare staff. Emphasize the comprehensive nature of data collection, covering both medical and non-medical cost factors.
  3. Cost Calculation: Clarify the methodologies used for calculating various costs, including direct medical costs, direct non-medical costs, and indirect costs. Discuss the approach used for assigning costs to specific resources.
  4. Statistical Analysis: Outline the statistical methods employed, such as the use of generalized linear models, to analyze cost data and identify predictors of cost variability.

Results

  1. Demographic and Clinical Characteristics: Present the demographic and clinical characteristics of the study population, highlighting any significant findings, such as the prevalence of certain complications.
  2. Cost Analysis: Detail the findings of the cost analysis, providing specific figures for different types of costs and discussing the distribution of these costs between the health system and households.
  3. Predictors of Cost Variability: Discuss the results related to predictors of cost variability, identifying factors such as gender, type of facility, and specific complications that significantly influence costs.
  4. Comparative Analysis: Compare your findings with existing literature, noting similarities and differences, and discuss possible reasons for any discrepancies.

Discussion

  1. Interpretation of Results: Provide a detailed interpretation of the results, emphasizing the significant economic burden on households and the implications for health policy.
  2. Policy Implications: Discuss how your findings can influence health policy, particularly in terms of financial protection for families and resource allocation for SAM treatment.
  3. Limitations: Acknowledge the limitations of your study, discussing how they might affect the interpretations and what future research could do to address these limitations.
  4. Future Research Directions: Suggest areas for further research, such as longitudinal studies to track the long-term economic impact of SAM treatment on households.

Conclusion

  1. Summary of Key Findings: Concisely summarize the main findings, reaffirming the economic impact of SAM treatment on households in Northern Senegal.
  2. Implications for Practice: Highlight the practical implications of your findings, advocating for policy changes to reduce the financial burden on affected families.
  3. Call to Action: End with a strong call to action, urging policymakers and health system stakeholders to consider the findings and implement strategies that will mitigate the economic impacts of SAM.
  4. Final Thought: Conclude with a reflective thought on the importance of integrating economic evaluations in the planning and implementation of health interventions, particularly in low-resource settings like Northern Senegal.
Comments on the Quality of English Language

A thorough English Language editing is recommended.

Author Response

(The authors gave the same response as above.)

Round 2

Reviewer 2 Report

Comments and Suggestions for Authors

The revisions made are satisfactory. The manuscript can be accepted for publication.